# Corrosion Resistance of Modified Hexagonal Boron Nitride (h-BN) Nanosheets Doped Acrylic Acid Coating on Hot-Dip Galvanized Steel

**DOI:** 10.3390/ma13102340

**Published:** 2020-05-19

**Authors:** Yongzhe Fan, Huazhen Yang, Haisheng Fan, Qi Liu, Chuang Lv, Xue Zhao, Mingxu Yang, Jianjun Wu, Xiaoming Cao

**Affiliations:** 1School of Materials Science and Engineering, Hebei University of Technology, 29 Guangrong Road, Tianjin 300132, China; fyz@hebut.edu.cn (Y.F.); yhz371502@126.com (H.Y.); 201611801001@stu.hebut.edu.cn (Q.L.); 15022330856@163.com (C.L.); hbgdwjj@hebut.edu.cn (J.W.); gd_sam@galvanize.com.cn (X.C.); 2Beijing Zhongke Pujin Special Material Technology Development Co., Ltd., 9 Zhongguancun South Avenue, Beijing 100081, China; fhs0830@163.com; 3Beijing Jitai Cold-Forging Technology Co., Ltd., 9 Zhongguancun South Avenue, Beijing 100081, China

**Keywords:** hot-dip galvanized steel, boron nitride nanosheet, corrosion resistance, acrylic coating

## Abstract

The hexagonal boron nitride (h-BN) nanosheets modified by silane coupling agent (KH560) were doped into acrylic acid coating on the surface of galvanized steel to improve its corrosion resistance. H-BN nanosheets modified by KH560 were prepared and characterised by scanning electron microscopy, transmission electron microscopy, atomic force microscopy, X-ray diffraction, and Fourier-transform infrared spectroscopy. The corrosion resistance of the acrylic acid coatings was measured by electrochemical testing. The results show that the corrosion current density of the coating with modified h-BN nanosheets was reduced from 2.2 × 10^−5^ A/cm^2^ to 2.3 × 10^−7^ A/cm^2^ compared with the acrylic acid coating. The impedance of the composite coating with modified h-BN is 4435 Ω·cm^2^, higher than the BNNS coating (2500 Ω·cm^2^) and the acrylic acid coating (1500 Ω·cm^2^). This is due to the physical barrier and electrical insulation properties of the hexagonal boron nitride (h-BN) nanosheets.

## 1. Introduction

Hot-dip galvanizing technology is often used to protect the steel substrate [1]. However, white rust easily accumulates on the surface of the galvanized layer in the atmosphere. This affects the appearance and quality of the hot-dip galvanizing products [2,3,4]. Passivation treatreams are usually applied for protecting the hot-dip galvanizing layer from corrosion. 

Chromium anticorrosion coating are widely used on hot-dip galvanized steel because of its excellent corrosion resistance [5,6,7,8], but chromium coating usually causes serious environmental pollution. It is also harmful to human health. Organic coatings are also used in hot-dip galvanized steel coating field owing to its better corrosion resistance. Organic compounds contained a large number of hydroxyl and carboxyl groups such as tannic acid and phytic acid [9,10]. These groups combined together with the galvanized layer through coordination bonds. In addition, when organo-silane is hydrolyzed, hydroxyl groups are formed, which are connected to the surface of the metal substrate by Si–O–Me bonds with good binding force [11,12,13,14]. A cross-linked silane coating can be prepared through another hydroxyl groups dehydration condensation on the surface of metal substrate. This silane coating can effectively isolate the water molecules to protect the metal matrix from corrosion. In addition, water-soluble acrylic and epoxy resins are also widely used in corrosion protection of hot-dip galvanized sheet [15,16,17]. However, organic coatings are brittle, prone to oxidation, and have a large number of micropores, which are produced during the coating process, so they need to be modified for better corrosion resistance [18,19]. 

Hexagonal boron nitride is widely used as a nanofiller for corrosion protection of metals because of its excellent chemical stability and insulation [20,21,22]. Boron nitride is chemically stable and insoluble in water, ethanol, and other solvents, which makes it an attractive anti-corrosion additive. However, its effectiveness is limited because it cannot completely disperse in the coating material. In recent years, the peeling of boron nitride into nanosheets has attracted widespread research attention. Compared with traditional boron nitride materials, boron nitride nanosheets (BNNS) have larger specific surface area, lower density, higher thermal stability, chemical stability, and corrosion resistance [23]. Currently, most research on this topic focuses on the addition of boron nitride to the coatings, but studies related to the addition of BNNS to waterborne acrylic coatings for the protection of hot-dip galvanized steel sheets are relatively scarce. BNNS are easy to aggregate in water [24]. BNNS need to be modified in order to improve the disperse property in waterborne coatings by surface modification. The modified BNNS retain good chemical stability, have hydroxyl groups attached to the surface, and can be uniformly dispersed in the waterborne acrylic coating. The modified boron nitride nanosheets are uniformly dispersed in the acrylic coating, so that the boron nitride nanosheets are evenly covered on the surface of the hot-dip galvanized sheet [25], thereby extending the service life of the hot-dip galvanized sheet. In addition, due to the dehydration condensation of KH560, a silane film is formed on the surface of boron nitride, so that the dried hot-dip galvanized sheet does not crack due to surface stress, and the silane film can further improve the corrosion resistance of the hot-dip galvanized sheet ability [26]. 

In the present work, the KH560 modified BNNSs were added into the preparation of the waterborne acrylic coating on the hot-dip galvanized steel sheet. The modified BNNSs were characterized by X-ray diffraction (XRD), fourier-transform infrared (FT-IR) spectra, X-ray photoelectron spectroscopy (XPS), and transmission electron microscopy (TEM). Microstructure and morphology of modified BNNSs coating was investigated by scanning electron microscopy (SEM). The corrosion resistance of modified BNNSs coatings was tested by an electrochemical workstation. This coating provided a physical shield to protect the metal substrate, and the corrosion resistance of the coating was effectively improved by the addition of the modified BNNSs.

## 2. Experimental

### 2.1. Materials

H-BN was purchased from Aladdin (Shanghai, China); dopamine hydrochloride were purchased from Kmart (Tianjin, China). Hydrogen peroxide and tris (hydroxymethyl aminomethane) (Tris buffer) were purchased from Xiensi (Tianjin, China). 3-glycidoxypropyltrimthoxysilane (KH-560) was purchased from Aladdin (Shanghai, China). Waterborne acrylic resin was obtained from Gongda Galvanising Company. Octylphenol polyoxyethylene ether (OP-10) was purchased from Tengyu Company. Polyoxyethylene ether (JFC) was obtained from Enochai Technology Company (Beijing, China). 1-hydroxyethane-1,1-diphosphonic acid (HEDP) was purchased from DaMao (Tianjin, China). Potassium permanganate was obtained from Tianjin Corus Fine Chemical Company, and concentrated sulfuric acid was purchased from TengJin Company (Tianjin, China). Hot-dip galvanized steel prepared for laboratory. All reagents used in this experiment were analytically pure and used without further purification. A commercially available hot-dip galvanized sheet was used in this work.

### 2.2. Stripping of BNNS

First, 1 g of boron nitride powder and 0.5 g of potassium permanganate powder were fully ground with a mortar and transferred into a cleaned flask. Then, 200 mL deionised water was added to the flask, and 30 mL of concentrated sulfuric acid was slowly added dropwise. The flask was kept in ice water at 0 °C for 12 h under continuous stirring. Subsequently, 20 mL hydrogen peroxide was added to the mixed solution. Finally, the obtained suspension was centrifuged at 3000 rpm for 10 min to remove large pieces of unpeeled boron nitride. The supernatant was then suction filtered, washed, and dried under vacuum at 40 °C for 48 h.

### 2.3. Preparation of Modified BNNS (560/BNNS)

BNNSs (0.6 g) and Tris buffer solution (0.16 g) were added into a 100 mL beaker containing ethanol (30 mL) and ultrasonicated for 30 min. Then, 0.24 g of dopamine hydrochloride were added to the mixed solution and stirred at 25 °C for 6 h to promote π–π bond polymerisation of BNNS and dopamine molecules. The modified BNNS obtained from this reaction were labelled as PDA/BNNS. One milliliter of KH-560 was added to the PDA/BNNS solution, and the mixed solution was kept at 60 °C for 5 h. After that, the solution was suction-filtered with a millipore filter. The obtained solid products of modified BNNS were washed several times with deionised water and ethanol, then dried in a vacuum drying box at 60 °C 24 h. These modified BNNS were named 560/BNNS.

### 2.4. Preparation of Anticorrosive Coatings

0.3 g of tannic acid were completely dissolved in a certain amount of deionised water with continuous stirring in a 100 mL beaker. Then, 0.1 g of JFC, 0.1 g of HEDP, and 0.1 g of OP-10 were added. After this solution was fully stirred, 40 g of waterborne acrylic solution was added, and finally, 0.02 g of modified BNNS were added to the above-mixed solution. The galvanised samples (using Q235 steel as raw material) were immersed in the above solution for 20 s and then taken out, washed with deionised water, and naturally dried [27]. This modified BNNS/waterborne acrylic-treated coating is labelled as 560/BNNS coating. Figure 1 and Figure 2 give a schematic of the synthesis of KH560/BNNS.

### 2.5. Characterisation Method

#### 2.5.1. Morphological and Structural Characterisations of BNNS and 560/BNNS

The crystal structures of BNNS and KH560 modified BNNS were characterised by X-ray diffraction (XRD) with Cu Kα radiation source at a scanning rate of 6°/min. The composition of the modified BNNSs was measured by X-ray Photoelectron Spectroscopy (XPS; ESCALAB250XI, Thermo Fisher Scientific, Waltham, MA, USA). Al K_α_ radiation was used with a spot size of 500 μm and an energy step of 0.100 eV. The thickness of the BNNSs were observed by atomic force microscopy (AFM; NT-MDT Prima, Saint Petersburg, Russia), while the micromorphology of the modified BNNSs was observed by transmission electron microscopy (TEM; FEI Tecnai G2 F20,Hillsboro, OR, USA,) performed at 200 kV. The coating morphology and microstructure of powdered boron nitride and peeled BNNSs were examined by scanning electron microscopy (SEM; Quanta 450 FEG, Hillsboro, OR, USA) at 20 kV. The modified BNNS was measured by X-ray photoelectron spectra (ESCALAB 250Xi, Thermo Fisher Scientific, Waltham, MA, USA) spectrometer. Fourier-transform infrared (FT-IR) spectra of the KBr pellets were recorded using an FT-IR spectrometer (TENSOR 27, Bruker, Billerica, MA, USA) at 2 cm^−1^ resolution.

#### 2.5.2. Characterisation of Corrosion Resistance

Electrochemical measurements were performed in a 3.5 wt.% aqueous NaCl solution using an electrochemical workstation (CHI-660E). A three-electrode electrochemical cell was used for all electrochemical measurements, where samples have an exposed area of approximately 1 cm^2^ used as a working electrode, a platinum electrode serving as a counter electrode, and a saturated calomel electrode serving as a reference electrode. When the fluctuation is less than 10 mV, the open circuit potential (OCP) was recorded; the test time of open circuit potential was 400 s. The potentiodynamic polarisation experiment was carried out with a scanning speed of 1 mv/s, and the potentiodynamic polarisation results were recorded from −0.3 V below the OCP to +0.3 V above the OCP. The corrosion potential (Ecorr) and corrosion current density (icorr) were obtained by Tafel extrapolation. Electrochemical impedance spectroscopy (EIS) is performed at scan frequencies ranging from 10^−2^ Hz to 10^5^ Hz. The EIS results were analysed using ZSimWin software (version 6.5).

## 3. Results and Discussion

### 3.1. Morphology and Composition of BNNSs and 560/BNNS

Figure 3 shows the XRD profiles of the h-BN, BNNSs and 560/BNNS. 

The diffraction peaks of the BNNS at 26.9°, 41.8°, and 44.0° correspond to the (002), (100), and (101) planes of h-BN. The intensity of the diffraction peak of (002) crystal plane of BNNS is higher than that of boron nitride, indicating that the crystal plane (002) of the boron nitride nanosheet is more exposed than that of the boron nitride powder, and it can be inferred that the nitrogen and boron stripped along the (002) crystal plane. The characteristic peaks of the BNNSs clearly shift toward lower 2θ angles, which indicates an increase spacing between crystal planes, based on Bragg’s equation: 2dsinθ = nλ [28]. These results indicate the absence of any intermediate during the preparation of the BNNSs and the successful peeling of BNNSs. The intensities of the diffraction peaks of dopamine and branched silane coupling agent-modified 560/BNNS decreased due to the coating of dopamine and silane coupling agent on the BNNS surfaces. Moreover, the grain size decreased and the crystallinity improved, further indicating successful modification of the BNNSs.

The morphology and thickness of the peeled BNNSs were observed by SEM and AFM, respectively. Figure 4a shows that the size of powder boron nitride is large and in an agglomerated state [29,30]. Figure 4b shows a successful strip of boron nitride to nanosheets, with a large number of layers peeled off. The morphology of these BNNSs was further characterized by TEM. Figure 4c also shows the morphology of BNNS. Figure 4d show the morphology of BNNS. Figure 4e show the AFM results reveal that the average thickness of BNNSs after peeling is 1.5 nm.

Figure 5 shows the FT-IR spectra of BNNSs and 560/BNNSs. 

For BNNSs, the characteristic absorption peaks appeared at 1380 cm^−1^ (B–N) and 800 cm^−1^ (B–N–B) (Figure 6a), whereas for modified BNNSs, the peaks at 2935 cm^−1^ and 2861 cm^−1^ originated from aliphatic CH_2_ bonding in dopamine hydrochloride (Figure 6b) [31,32]. In addition, new characteristic peaks appeared at 1715 cm^−1^ (C=C stretching), 1194 cm^–1^ (Si–C–R bonding), 1100 cm^−1^ (Si–O–Si bonding), and 1018 cm^−1^ (Si–O–C bonding). Moreover, a particularly noteworthy peak appears at 879 cm^–1^, which originates from the vibrations of epoxy groups. These results prove that the silane coupling agent (KH-560) and dopamine hydrochloride are successfully attached on the BNNS surface.

XPS results of BNNS and 560/BNNS are shown in Figure 6. All the spectra were calibrated with respect to the C 1s binding energy at 284.7 eV as a reference. The XPS survey of BNNS and 560/BNNS is shown in Figure 6a,b. The presence of Si is due to the addition of KH-560. It can be seen from Figure 6c,d that the binding energies of N1s and B1s are 397.9 eV and 190.5eV, respectively. The high-resolution C 1s and Si 2p spectra of modified BNNSs are shown in Figure 6e,f. The presence of C, O, B, N, and Si elements in the 560/BNNSs are evident. Figure 6e shows the presence of C–Si (284.6 eV), C–C (284. 8 eV), C–N (285.5 eV), and C–O (286.5 eV). Furthermore, the peaks at 101.6 eV and 102.4 eV in the Si 2p region correspond to the Si–OH and hydrochloride of KH-560, respectively. These results suggest that boron nitride was modified by dopamine hydrochloride and KH-560 [33,34].

The 560/BNNS was analyzed by TEM [35]. It can be seen from Figure 7a that a thin film is observed on the boron nitride nanosheet modified by the silane coupling agent. In addition, Figure 7b shows a film, which is a film layer of about 5 nm formed by KH560 chemical bonding with dopamine on the surface of boron nitride, and KH560 dehydration condensation.

### 3.2. Microstructure and Morphology of Anticorrosive Coatings

The microstructures and morphologies of waterborne acrylic coating, BNNS/waterborne acrylic coating, and modified BNNS/waterborne acrylic coating were analysed by SEM. As shown in Figure 8a, the morphology of the waterborne acrylic coating has many cracks, which reduces its effectiveness in protecting the encased material from corrosion and leads to the electrochemical corrosion of the galvanised steel. Figure 8c shows a SEM image of modified BNNS/waterborne acrylic coating. The width of the crack is smaller than that of waterborne acrylic coating because BNNSs can promote molecular cross-linking of acrylic acid during film formation and reduce cracking caused by excessive evaporation of water molecules during drying , thus providing effective protection against corrosion to the galvanished layer [36,37,38]. Figure 8e shows the morphology of modified BNNS/waterborne acrylic coating. Compared with the waterborne acrylic coating and BNNS/waterborne acrylic coating, the surface of modified BNNS/waterborne acrylic coating is smooth and crack-free. This is because the silane coupling agent (KH-560) on the surface of boron nitride can connect with acrylic acid [39,40]. This connection enhances the strength and reduces the number of cracks in the coating, thus protecting the galvanised layer and making it less susceptible to corrosion. Figure 8b,d,f show the cross sectional microscopic images of different coating. The thickness of each coating is approximately 2 μm.

### 3.3. Electrochemical Studies

Tafel polarization analysis to calculate the corrosion rate of the coating. The corrosion current density (i_corr_) and corrosion rate are the two most important parameters and are usually used to evaluate the speed of corrosion. The formula for calculating the corrosion rate is: (1)$$v_{corr}=EK\frac{i_{corr}}{\rho }$$
where *E* is the equivalent weight of zinc (EZinc = 1.22 g/A·h), *K* is the coefficient of equivalent weight (K = 0.0876 in this formula), icorr is the corrosion current density, and ρ is the density of zinc (ρ = 7.14 g/cm^−3^).

Figure 9 and Table 1 show the potentiodynamic polarization curves of different coatings. Results reveals that there are significant differences in anticorrosion performance. 

Pure waterborne acrylic coating has the lowest corrosion potential of −1.13 V, with a corrosion current density of 2.2 × 10^−5^ A/cm^2^. The corrosion potential of the BNNS coating is −1.06V and the corrosion current density is 9.7 × 10^−6^ A/cm^2^. The corrosion potential of the coating containing 560/BNNS reaches −1.02 V, with a density of the corrosion current density of 2.3 × 10^−7^ A/cm^2^. The corrosion current density is reduced at two orders of magnitude. Although the corrosion rate of pure waterborne acrylic coating is not as high as 3.29 × 10^−1^ mm/year, the addition of BNNSs further reduces the corrosion rate by an order of magnitude to 1.45 × 10^−2^ mm/year and improves the dispersibility of modified boron nitride in water-soluble acrylic acid and the corrosion rate of the coating further reduces to 3.44 × 10^−3^ mm/year, which is two orders of magnitude lower than that of pure waterborne acrylic coating. The BNNS and 560/BNNS is wider than that of pure waterborne acrylic treated coatings the corrosion protection areas. This shows that BNNSs and modified BNNSs improve the corrosion resistance of coating. 

The anticorrosive properties of acrylic coatings were studied by EIS testing. All polymers are infiltrated by potentially corrosive substances [41]. Thus, in the initial stage of corrosion, the electrolyte penetrated the coating through the inherent micropores, but the surface of the galvanised sheet does not corrode. However, the corrosion products eventually accumulate at the zinc/coating interface [42]. If the waterborne acrylic acid coating contains too many pores, the above conversion process will occur over a short time, and the excellent anti-corrosion coating will last longer during the corrosion process. 

Figure 10a–c shows the Nyquist and Bode plots of pure waterborne acrylic coating, BNNS/waterborne acrylic coating, and 560 BNNS/waterborne acrylic coating after immersion in 3.5% NaCl aqueous solution for 1 h. The radius of the impedance arc of the pure waterborne acrylic coating is smaller than that of the other two coatings (BNNS/waterborne acrylic coating and modified BNNS/waterborne acrylic coating); where the radius of the arc represents the magnitude of impedance, the larger the arc length diameter, the better the anticorrosive effect of the coating. The impedance of the 560/BNNS composite coating is 4435 Ω·cm^2^, higher than the BNNS coating (2500 Ω·cm^2^) and the pure waterborne acrylic coating (1500 Ω·cm^2^). The resistance of 560/BNNS coating is 2.95 times higher than that of pure waterborne acrylic coating. Additionally, the peak of the phase angle of the 560/BNNS coating is higher than pure waterborne acrylic coating. Porosity is also an important index for evaluating the corrosion resistance of coatings. The porosity (P) is calculated according to the following formula:*P* = 2.3 × (*R_p_*_0_/*R_p_*) × *exp.* (△*E_corr_*/*β_a_**)*(2)
where *R_p0_* and *R_p_* are the polarisation resistances of the untreated sample and the coating sample, respectively, △*E_corr_* is the corrosion potential difference between the untreated sample and the corrosion-treated sample, and *β_a_* is the anode taffy of the untreated sample Seoul slope [43]. The pure acrylic coating had the largest porosity, followed by the BNNS coating, and the 560/BNNS coating. This result is highly consistent with the SEM image mentioned above in Figure 8. These results verify that the added 560/BNNS nanocomposite coating exhibited superior corrosion resistance over pure coating and BNNS coating, owing to its good dispersion and the excellent inherent plugging of micropores. 

EIS analyses suggest that an equivalent electric circuit model shown in Figure 10d is used to fit the results. The fitted results represent the state of corrosion in the coating, where R_s_ is the solution resistance, R_pore_ is the resistance in the pore of the coating defects, Qc and Q_dl_ is the constant phase angle element representing capacitive properties of the coating, Rt is the charge transfer resistance of the substrate/electrolyte, and R_corr_ and Q_corr_ represent the resistance and capacitance of the corrosion products, respectively [44]. The following Table 2 lists the fitting parameters of the initial corrosion.

As shown in Figure 11a, the impedance modulus of the pure waterborne acrylic coating indicates its poor corrosion resistance compared to the other two coatings. 

This is because the corrosive solution penetrates the coating and corrodes the metal substrate. After long-term immersion, the coating containing BNNSs exhibited better corrosion resistance than the pure waterborne acrylic coating. However, due to the agglomeration of BNNSs, it could not uniformly disperse in the coating solution, which resulted in a less effective anti-corrosion effect. The modified 560/BNNS coating acted as a good anti-corrosion barrier for the metal matrix surface because it could be uniformly dispersed in the solution. Hence, the modified 560/BNNS coating greatly improved the corrosion resistance as compared with the other two coatings. Table 3 shows the electrochemical fitting parameters for long-term immersion. By comparing the experimental data in Table 2 and Table 3, it can be seen that the 560/BNNS coating can better protect the hot-dip galvanized steel plate from corrosion under the conditions of humidity and high salt.

### 3.4. Mechanism of Corrosion Resistance Enhancement

The addition of 560/BNNS can improve the corrosion resistance of waterborne acrylic coatings because the nanosheets are uniformly distributed in the waterborne acrylic coating and promote mutual crosslinking of organic molecules. In addition, due to the impermeability of 560/BNNSs, the coating is evenly and smoothly. Crosslinking through silane membranes can effectively control the propagation of cracks. These results show that the cracks in the organic coatings decreased considerably during drying. Due to these effects, a relatively complete and dense coating can be obtained, which enhanced its barrier performance. This can be observed by the morphology of BNNS/waterborne acrylic and 560/BNNS/waterborne acrylic coatings. Another reason for the improved corrosion protection is shown in Figure 12. By adding BNNS and 560/BNNS to the waterborne acrylic solution, the ion exchange between the solution and the metal substrate can be effectively suppressed, and the diffusion path of the ions is lengthened. This leads to a reduction in the corrosion rate and an increase in the corrosion resistance of the composite coating.

## 4. Conclusions

KH-560-modified BNNS was successfully synthesized by dopamine binding to boron nitride through a π–π bond, and the chemical interaction between the catechol group in dopamine and KH560. The modification of BNNSs reduced the agglomeration of the nanosheets and promoted the formation of a hybrid material. The corrosion performance of the composite coating is enhanced. The EIS results show that the coating containing 560/BNNS can improve the anticorrosion performance of coating compared with pure coating. The modified 560/BNNS was used as filler for an aqueous acrylic acid coating; BNNSs has good physical barrier performance. KH560 acts as a connecting bridge that improves the hydrophobicity of the resin film and the adhesion of the coating to the substrate. Due to the effect of silane, cracks in the coating are significantly reduced and the coating surface is smoother. Based on the above experimental data, the modified boron nitride nanosheets dispersed in acrylic acid are very effective for the corrosion protection of hot-dip galvanized sheet.

## Figures and Tables

**Figure 1 materials-13-02340-f001:**
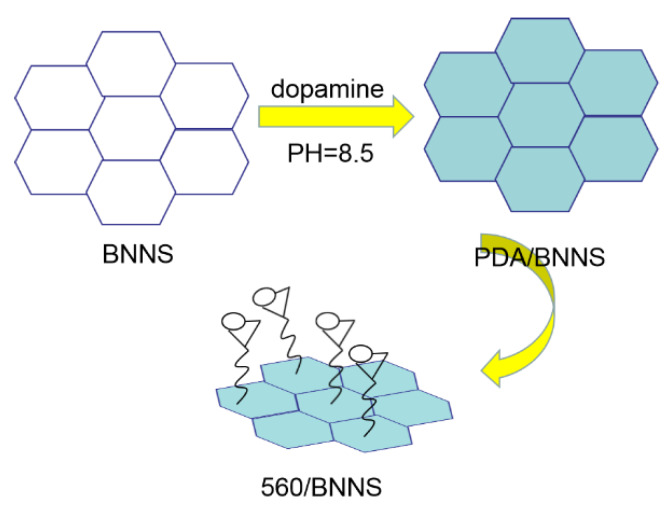
Schematic diagram of KH560 modified boron nitride nanosheets.

**Figure 2 materials-13-02340-f002:**
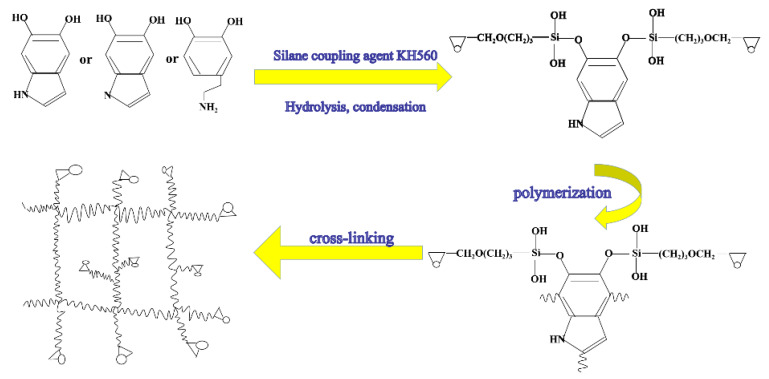
Schematic diagram of modified boron nitride reaction.

**Figure 3 materials-13-02340-f003:**
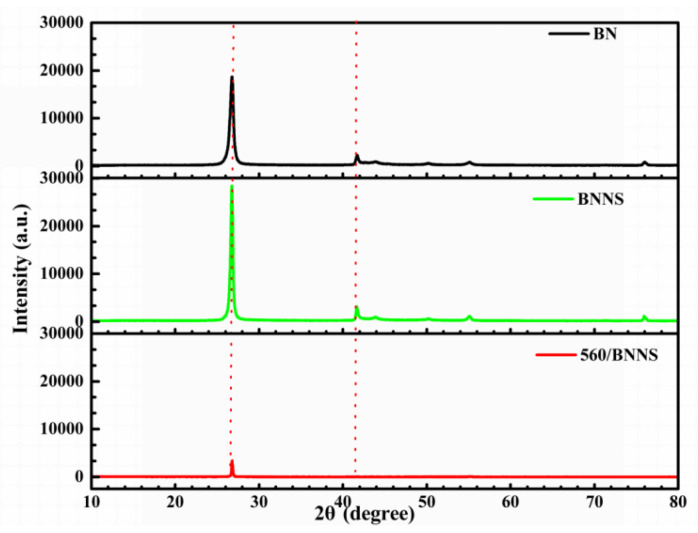
XRD result of h-BN, BNNS and 560/BNNS.

**Figure 4 materials-13-02340-f004:**
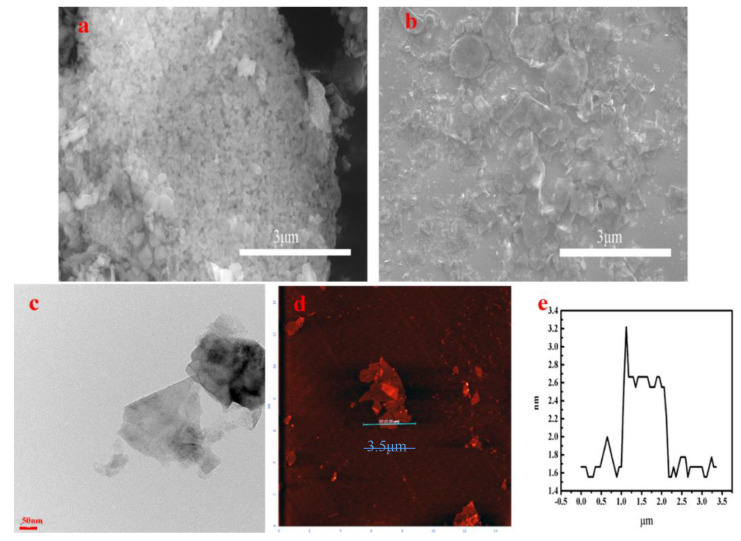
The SEM images of h-BN (**a**) and BNNS (**b**), (**c**) is the TEM image of BNNS, (**d**) are AFM result of BNNS, (**e**) is the thickness of BNNS.

**Figure 5 materials-13-02340-f005:**
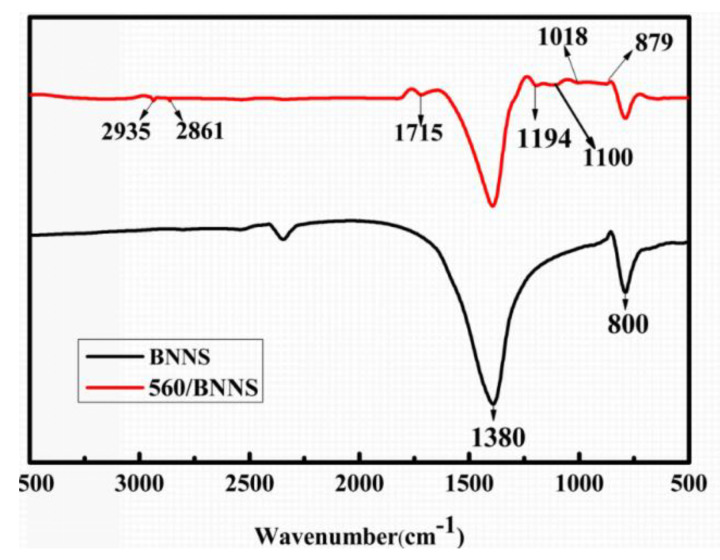
FT-IR result of BNNS and 560/BNNS.

**Figure 6 materials-13-02340-f006:**
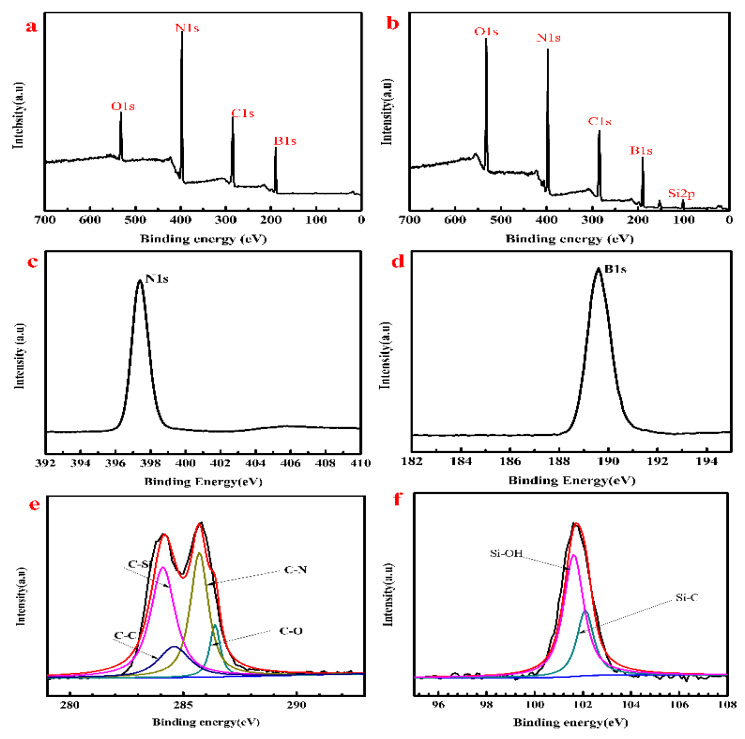
(**a**,**b**) are survey spectra of XPS results of BNNS and 560/BNNS, high resolution spectra (**c**,**d**) of BNNS, high resolution spectra (**e**,**f**) of 560/BNNS.

**Figure 7 materials-13-02340-f007:**
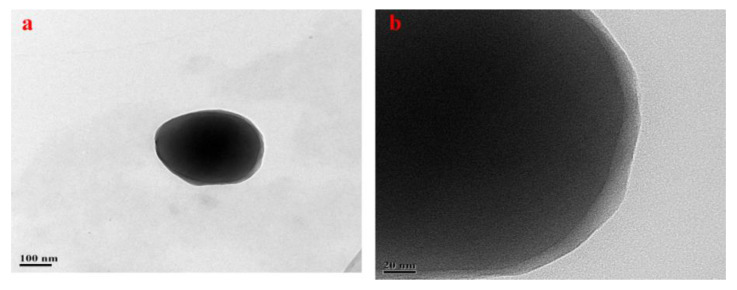
TEM images (**a**,**b**) of 560/BNNS.

**Figure 8 materials-13-02340-f008:**
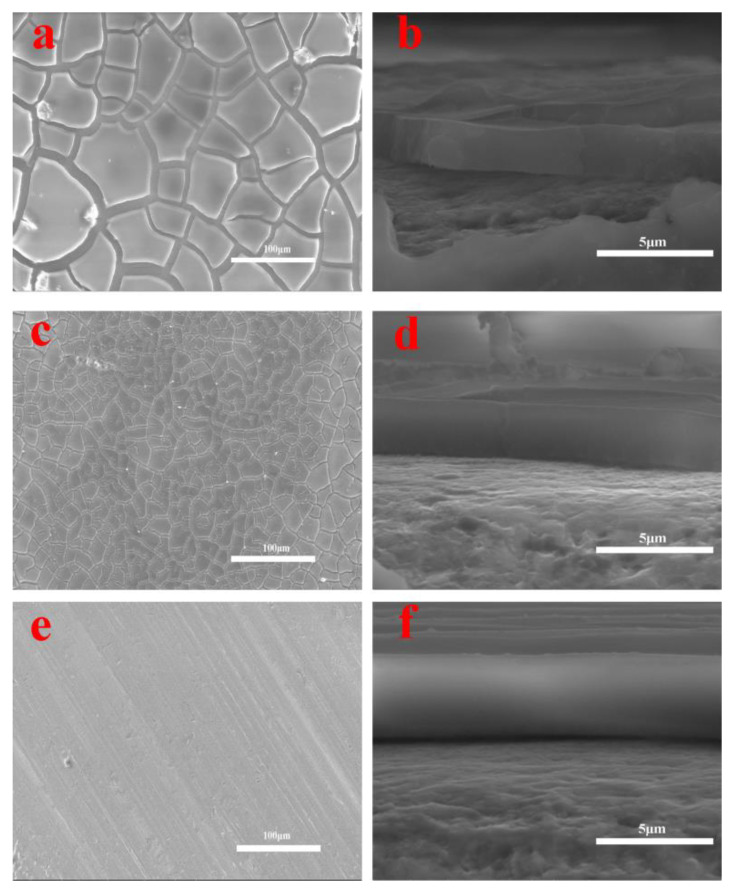
SEM results of waterborne acrylic coating, BNNS/waterborne acrylic coating and 560/BNNS waterborne acrylic coating morphologies. (**a**) The waterborne acrylic coating; (**b**) the cross section of waterborne acrylic coating; (**c**) the BNNS/waterborne acrylic coating; (**d**) the cross section of BNNS/waterborne acrylic coating; (**e**) the 560/BNNS waterborne acrylic coating; (**f**) the cross section of 560/BNNS waterborne acrylic coating.

**Figure 9 materials-13-02340-f009:**
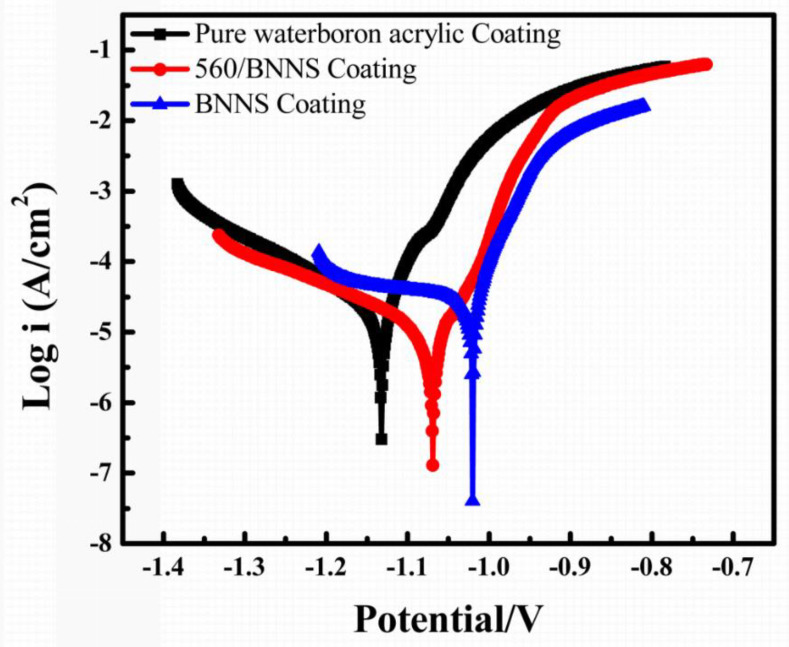
Potentiodynamic polarization curves of pure waterboron acrylic coating, BNNS coating and 560/BNNS coating.

**Figure 10 materials-13-02340-f010:**
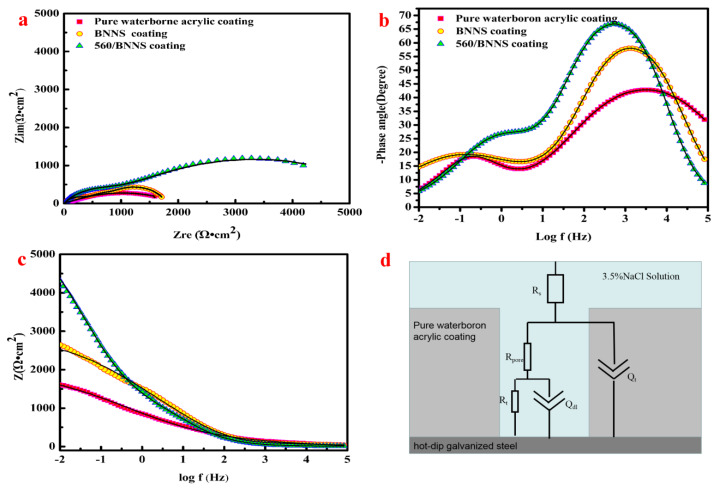
The EIS results of pure waterborne acrylic coating, BNNS coating and 560/BNNS coating. (**a**) is the Nyquist plot, (**b**,**c**) are the Bode plots, (**d**) is the equivalent electric circuit.

**Figure 11 materials-13-02340-f011:**
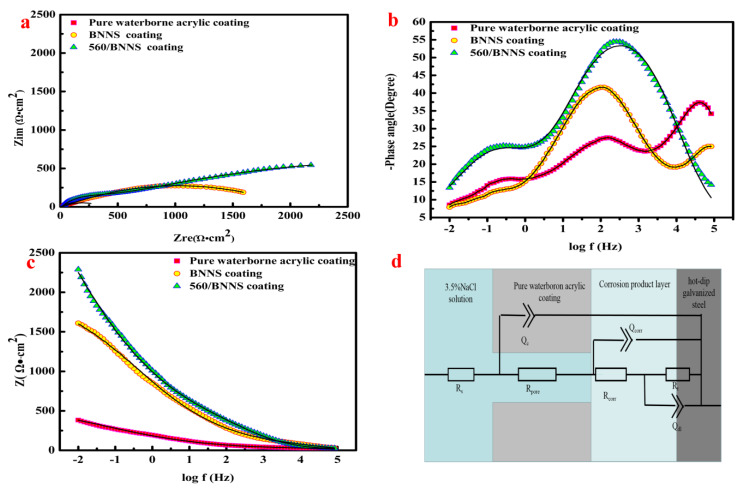
The EIS results of pure waterborne acrylic coating, BNNS coating and 560/BNNS coating after long-term immersion. (**a**) is the Nyquist plot, (**b**,**c**) are the Bode plots, (**d**) is the equivalent electric circuit.

**Figure 12 materials-13-02340-f012:**
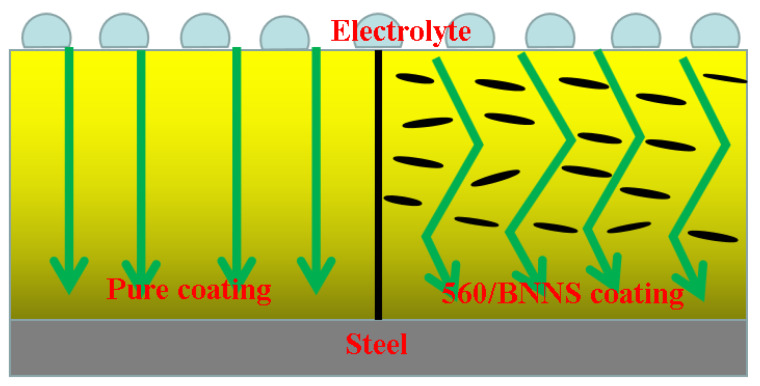
Corrosion resistance mechanism model diagram.

**Table 1 materials-13-02340-t001:** Potentiodynamic polarization parameters and corrosion rate calculation results.

Sample	I_corr_ (A/cm^2^)	E_corr_ (V)	Corrosion Rate (mm/year)
Pure waterboron acrylic Coating	2.2 × 10^−5^	−1.13	1.432 × 10^2^
BNNS Coating	9.7 × 10^−6^	−1.02	1.347 × 10^1^
560/BNNS Coating	2.3 × 10^−7^	−1.06	3.734 × 10^0^

**Table 2 materials-13-02340-t002:** EIS fitting parameters of various samples.

Sample	Pure Waterborne Acrylic Coating	BNNS Coating	560/BNNS Coating
R_s_ (Ω·cm^2^)	5.32	6.14	7.23
Q_c_ (Ω^−1^·cm^−^^2^·s^n^)	1.32 × 10^−^^4^	5.37 × 10^−^^5^	1.44 × 10^−^^6^
R_pores_ (Ω·cm^−2^)	172	406	958
R_t_ (Ω·cm^−2^)	1500	2500	4435
Q_dl_ (Ω^−1^·cm^−^^2^·s^n^)	6.52 × 10^−^^3^	2.65 × 10^−^^3^	5.84 × 10^−^^4^

**Table 3 materials-13-02340-t003:** EIS fitting parameters of long term immersion.

Sample	Pure Waterborne Acrylic Coating	BNNS Coating	560/BNNS Coating
R_s_ (Ω·cm^−2^)	3.97	5.23	5.46
Q_c_ (Ω^−1^·cm^−^^2^·s^n^)	3.24 × 10^−3^	4.21 × 10^−4^	3.16 × 10^−5^
R_pore_ (Ω·cm^2^)	26.8	254.2	485.3
Q_dl_ (Ω^−1^·cm^−^^2^·s^n^)	1.24 × 10^−^^3^	1.42 × 10^−^^3^	2.54 × 10^−4^
R_t_ (Ω·cm^−2^)	469	1538	2525
R_corr_ (Ω·cm^−2^)	248.5	176.3	122.1
Q_corr_ (Ω^−1^·cm^−^^2^·s^n^)	5.87 × 10^−^^3^	4.57 × 10^−^^3^	5.74 × 10^−3^

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
