# Peer review of "Corrosion Resistance of Modified Hexagonal Boron Nitride (h-BN) Nanosheets Doped Acrylic Acid Coating on Hot-Dip Galvanized Steel"

_materials, 2020, doi:10.3390/ma13102340_

Round 1

Reviewer 1 Report

The acrylic coating modification proposed in this work is interesting. The research methodology used is also correct. However, the main theme - corrosion resistance - is the weakest side of this work. The entire section devoted to the corrosion resistance is inconsistent, incomplete and there is a lack of information. In addition, it is worth mentioning about minor and major errors such as:

  • too high precision of the results from polarization measurements (jcorr 2.279e-5 A/cm2 - just 2.3, and even 2) - Tafel extrapolation of unstable system (after 1 hour exposure!) is without sense, and bring large error,
  • lack of experimental details in section 2.5.2,
  • Figure 4 is difficult to read,
  • the absence of y-axis label in Fig. 5 (normalized scale or not?),
  • line 217 - first sentence is unclear,
  • Fig. 9 - "Log (i/A)" on y-axis. Where is log scale? It is decimal,
  • Table 1 - icorr for 560/BNNS Coating is 10-7. From Fig. 9 it should be in the same order of magnitude as for the other samples. But it does not. Why?
  • Line 229 - The Authors write about corrosion rate of the coating. From Table 1 the values for CR are from 3.7 to 143 mm/year (!?). To which material do these values correspod? Rather not for zinc?
  • Lines 257-258 - Formula (1) - Where did the Authors get such an equation? Reference source is missing.
  • Lines 252-253 - The Authors write about impedance and resistance alternatively. It is not correct.
  • Figs 10 and 11 - Similar shape of the spectra, two time constants, but two different electric circuits... How did the Authors obtain a good fit with the model from Fig. 10d to the spectra from Fig. 10b? If so, then the same model should be used for the spectra from Figure 11b... It is impossible. Spectra for 560/BNNS from both figures (both immersion times) are similar - two-time constant model should be used for them.
  • Tables 2 and 3 - The resistances from these tables do not correspond to these visible in Bode plots. They differ in orders of magnitude and values. The whole gives the impression of inconsistency.

Author Response

Dear teacher:

I'm sorry to get back to you now, because of the outbreak in China, the students have been home and haven't been able to return to school. I always wanted to add some experimental data, but the school has not been open until now. I made some changes to the data.

First, I modified the accuracy of the polarization curve data.

Second, add 2.5.2 experimental details。

Third, modify Figure 4 and  figure 9

Teacher, Figure 5 generally does not add the Y coordinate. I consulted other documents and did not add them, so I did not modify them.

Modify line 217

The experimental samples in Table 1 and Figure 9 are the same batch of samples, but the sample numbers are different, so the data has errors。

 Formula (1) add references。

Corrosion rate for line 229 is coating instead of zinc。

Modification of lines 252-253

EIS data modified accordingly。

Reviewer 2 Report

The authors presents the synthesis of modified hexagonal boron
3 nitride nanosheets doped acrylic acid and the use of it for improve the corrosion resistance of galvanized steel.

The paper could be published after minor observations:

The material characteristics of the galvanized steel samples should be provided.

Also, the surface characterization and electrochemical studies (EIS and Tafel) for galvanized steel should be given instead of pure zinc.

Please check the units of measure: e.g. mV instead of mv (page 4 lines 132 and 134)

The English should be carefully checked.

Author Response

Dear teacher 

I'm sorry to reply to your email now. It was postponed until now because of some experimental data.

I have made some changes to the article, as shown below。

First, I only added the steel material model because the hot-dip galvanized samples were prepared in the laboratory.

Second, because of the epidemic situation in China, students have been staying at home and are not allowed to return to school. They cannot supplement the EIS data of the corresponding pure zinc samples. I hope the teacher forgive me。

Modify writing details such as mV.

Reviewer 3 Report

Thank you for submitting the draft paper to Materials. The addition of hBN to paints is an interesting concept and this has been reported previously. In fact, a paper was published in Corrosion Science in 2018 (https://doi.org/10.1016/j.corsci.2017.11.022) which was not reported in the literature review in this paper. Further, tjese are other papers in the area which are relevant such as ACS Appl. Mater. Interfaces 2013, 5, 10, 4129-4135 which needs to be referenced. The work is interesting and the researchers would be interested in this. However, this needs further work to get the manuscript to a stage where this could be taken further. Please see the details in the attached pdf. In Summary, the following changes are required:

  1. The level of English in the introduction needs improvement. In fact, re-writing the introduction would help the reader understand the purpose of the paper and how this paper is trying to address gaps in knowledge.
  2. There are some sentences which are factually incorrect. These need to be corrected. For example, in line 42 "...organic coatings are brittle..." This is a very generic statement. This needs to be more specific or clarified that some of them can become embrittled during service due to UV degradation...
  3. Some figures need editing. The scales  are not easy to read in Figure 4. Figure caption used in Figure 8 needs rewording. I have provided some suggestions.
  4. The currently used paint systems provide better protection to steel than the system used here. This needs to be highlighted (line 226-227)
  5. A simple equivalent circuit has been used. Please explain the rationale. I am not sure why the solution resistance would be different if you have used the same electrolyte.
  6. The use of words in line 296 needs some attention

Author Response

Dear teacher 

I'm sorry to send you a message now. Because of the epidemic situation in China, students have been staying at home and cannot return to school for experiments and supplement the experimental data. Therefore, it has been postponed until now. I hope the teacher will understand.

Written in English, I made some changes.

Figures 4 and 8 have been modified.

Teacher, the solution resistance is different because during the experiment, with the extension of the experiment time, the water in the solution decreases, and the solution concentration will be different, resulting in a certain deviation in the solution resistance. This is a normal phenomenon and belongs to experimental error.

Round 2

Reviewer 2 Report

The required experiments would increase the quality of the paper. It would help to rationalize the  obtained results.

Author Response

Dear teacher

Teacher, I added the data of pure waterborne acrylic coating to the manuscript. Before, because my expression in the manuscript was unclear, it caused some troubles to the teacher. I am deeply sorry。

Reviewer 3 Report

Thanks you for making some of the changes requested. However, the changes are not adequate. Please check the figures captions and make them legible.

Figure 4: The scale bar is not visible

Figure 6: The quality of the figures is poor

Figure 7: The caption and figure appear separately

Table 1: The significant digits need to be taken into account when reporting the potential in V

Figure 10: The scale bar, scale and the text are not legible

Figure 11: The scale bar, scale and the text are not legible

Author Response

Dear teacher

I have modified the pictures in the manuscript, please ask the teacher for review. If there is a lack of modification, please let me know.

Round 3

Reviewer 2 Report

The paper could be accepted.

Author Response

thank you